# Cutting out Cholecystectomy on Index Hospitalization Leads to Increased Readmission Rates, Morbidity, Mortality and Cost

**DOI:** 10.3390/diseases9040089

**Published:** 2021-12-06

**Authors:** Karthik Gangu, Aniesh Bobba, Harleen Kaur Chela, Omer Basar, Robert W. Min, Veysel Tahan, Ebubekir Daglilar

**Affiliations:** 1Division of Hospital Medicine, Department of Medicine, University of Missouri, Columbia, MO 65212, USA; karthik.gangu@health.missouri.edu; 2Division of Hospital Medicine, Department of Medicine, John H Stroger Hospital of Cook County, Chicago, IL 60612, USA; anieshbobba@gmail.com; 3Division of Gastroenterology and Hepatology, University of Missouri, Columbia, MO 65212, USA; chelah@health.missouri.edu (H.K.C.); obasar@health.missouri.edu (O.B.); 4Department of Medicine, Rush Medical Collage, Chicago, IL 60612, USA; robert_w_min@rush.edu

**Keywords:** cholecystectomy, gallbladder, gallstone, pancreatitis, mortality, morbidity, readmissions

## Abstract

Biliary tract diseases that are not adequately treated on index hospitalization are linked to worse outcomes, including high readmission rates. Delays in care for conditions such as choledocholithiasis, gallstone pancreatitis, and cholecystitis often occur due to multiple reasons, and this delay is under-appreciated as a source of morbidity and mortality. Our study is based on the latest Nationwide Readmissions Database review and evaluated the effects of postponing definitive management to a subsequent visit. The study shows a higher 30-day readmission rate in addition to increased mortality rate, intubation rate, vasopressor use in this patient population and significantly added financial burden.

## 1. Introduction

Cholecystitis, choledocholithiasis and acute biliary pancreatitis (acute gallstone-related diseases) represent a spectrum of biliary diseases that warrant evaluation for cholecystectomy on index hospitalization. In cases of mild biliary pancreatitis cholecystectomy does not need to be delayed, whereas in cases of severe disease an interval cholecystectomy once inflammation resolves may be necessary [1]. Most cases of acute pancreatitis result from alcohol or a biliary source, either due to gallstones or sludge [2,3]. Several metanalysis showed interval cholecystectomy after mild biliary pancreatitis is associated with high risk of readmission for recurrent biliary events [4,5,6]. Prospective randomized studies showed that index admission laparoscopic cholecystectomy in mild to moderate acute biliary pancreatitis reduced the risk for recurrent attacks and was not linked to increased operative challenges or morbidity in the peri-operative period [7,8,9,10]. Hence removal of the source via cholecystectomy is important to prevent recurrence especially as recurrent attacks can be associated with worse outcomes. For cases of acute cholecystitis, an early laparoscopic cholecystectomy has been associated with reduced length of hospitalization and no significant perioperative complications [11,12,13,14,15,16,17]. Similar findings are found in biliary colic and choledocholithiasis [18,19,20].

Despite this the decision to pursue cholecystectomy on the initial hospitalization should consider patient specific characteristics and should be an individualized decision so that it can be performed safely. Current guidelines recommend that diseases such as choledocholithiasis, gallstone pancreatitis and cholecystitis should ideally be addressed on initial/index hospitalization or early within 2–4 weeks by performing cholecystectomy [21,22]. This is in light of endoscopic retrograde cholangiopancreatography (ERCP) being on the rise in popularity and the decline of surgical common bile duct (CBD) exploration [23,24], with outcomes studied demonstrating same session laparoscopic cholecystectomy with intraoperative ERCP having the most successful, safe and short length of hospital stay (LOS) [25].

Although the importance of early cholecystectomy has been emphasized especially on index hospitalization to prevent complications. Yet, the significant impact on patient’s readmission and the monetary costs of readmission, especially with choledocholithiasis, gallstone pancreatitis and cholecystitis and the independent predictors of readmission based on socioeconomic factors have been an understudied intersectionality of patient populations in the United States. Another aspect of delaying cholecystectomy that needs to be recognized is that those who are discharged on the index hospitalization without undergoing cholecystectomy may not do so in timely manner post discharge. Hence, patients who are readmitted within 30 days often tend to be readmitted as a consequence of not undergoing outpatient cholecystectomy. We used the National Readmission Database to evaluate the impact and burden on the US healthcare system of not performing a timely cholecystectomy on the index hospitalization in addition to its direct impact on morbidity and mortality.

## 2. Methods

Data source: We conducted retrospective analysis utilizing the largest publicly available all-payer NRD for the year 2018. NRD can be obtained from the healthcare cost and utilization project (HCUP), which Agency sponsors for Healthcare Research and Quality (AHRQ) [26]. Unweighted 2018 NRD contains approximately 18 million discharges for that year, while weighted sample estimates around 35 million discharges in the US. NRD draws its sample from 28 states, representing 59.7% of the total US population and 58.7% of all US hospitalizations.

Study population: 

We included patients with age ≥18 yrs and non-elective admission with a principal diagnosis of cholelithiasis or acute biliary pancreatitis. Patients who underwent open or laparoscopic cholecystectomy during index hospitalization were excluded from the study. December month admissions were excluded as data regarding 30-day readmission cannot be calculated. ICD-10 Clinical Modification (CM) and ICD-10 procedure codes were utilized to identify the patient sample.

Variable selection: Variables included in the study were divided into three categories:Patient level: Age, sex, median household income in the zip code, insurance status and Charlson comorbidity score.The severity of illness: Mechanical ventilation, vasopressor use, length of stay and disposition.Hospital level: Hospital location, teaching status and bed size.

Study Outcomes: Primary outcome was all-cause 30-day readmission. Any non-traumatic admission within 30 days of discharge after index admission was considered as readmission. Secondary outcomes were (a) in-hospital mortality for index and readmissions, (b) 30-day mortality rate following index admissions, (c) top 10 principal diagnosis in readmitted patients, (d) total length of stay and resource utilization associated with readmissions and (e) independent predictors of readmission.

Statistical analysis: All statistical analysis were conducted utilizing STATA 17.0 version (STATA CORP, College Station, TX, USA). Weights provided in the NRD dataset were applied for all analysis. All *p*-values were two-sided, and 0.05 was considered statistically significant. In index admission and readmission, categorical variables were compared using the Chi-square test, and continuous variables were compared using linear regression. Univariate Cox regression was used to calculate the unadjusted hazard ratio. Multivariate Cox regression model to identify independent predictors.

## 3. Results

Patient characteristics: 

A total of 93,140 patients were included in the study after excluding patients as described in Figure 1. Patient and hospital-level characteristics for index hospitalization are shown in Table 1. Females comprised 56.44%, with a mean age of 61.6 (SD 21.8). A total of 52.48% of the population were aged ≥65 yrs. Around 71.35% had either Medicare or Medicaid. One-fourth of the patients had nonroutine discharge (including discharges to a skilled nursing facility or home health or left against medical advice). More than half of the patients were treated in large hospitals.

30-day all-cause readmission:

Of 93,140 patients, 922,53 were discharged alive of which 11,292 (12.24%) were readmitted in 30 days. Figure 2 shows the Kaplan–Meier survival curve.

Comparison of all outcomes for index admission vs. readmission:

A total of 887 (0.95%) died during index hospitalization, while 288 (2.57%) died during readmission (0.95% vs. 2.57%, *p* < 0.001). Out of total 887 patients who died during index hospitalization, 694 (78.2%) of them were age ≥65 years-old. Mechanical ventilation, vasopressor use and total costs were significantly higher in readmission (Table 2). 

Most common reason for readmission: 

The most common cause of readmission was sepsis secondary to biliary source, and nine out of ten reasons for readmission were due to gall stones-related diseases. Table 3 shows the ten most common reasons for readmissions.

Resource utilization due to readmissions: 

Mean LOS for readmitted patients was 5.16 days when compared to index admission at 4.18 days (*p* < 0.001). Mean hospitalization charges in readmitted vs. index admission were 61,038$ vs. 46,078$ (*p* = 0.01). Total LOS incurred due to readmission was 57,800 days, with resulting total hospitalization charges of $682 million.

Illness severity: 

Readmitted patients had higher mechanical ventilation (3.64% vs. 1.84% *p* < 0.001), vasopressor use (0.94% vs. 0.29% *p* < 0.001) and non-routine discharge (35.28% vs. 25.49% *p* < 0.001).

Independent predictors of readmission: Variable selection, multivariate Cox regression model building was described in the methods section. Independent predictors for 30-day readmission were higher Charlson comorbidity score, younger age (<45 yrs), Medicare and Medicaid insurance, non-routine discharge and longer length of stay. Predictors associated with decreased 30-day readmission were the older age group (≥65 yrs), private insurance, female sex and routine discharge. The rest of the variables had no bearing on the 30-day readmission rate, as shown in Table 4.

## 4. Discussion

This study shows the financial and economic burden that stems from readmissions resulting from not performing cholecystectomy on the index hospitalization and the high morbidity and mortality associated with it. The financial burden of readmission is reflected in the resource utilization and the added length of stay in the hospital. Of those discharged after index hospitalization without having received cholecystectomy, within the parameters of our study described in Figure 1, 12.24% are readmitted. The readmitted population alone creates a patient population that is redundant. The burden of total readmission length of stay is roughly 57,800 days, with a combined financial burden of $682 million in 2018. In readmitting this patient population, there is a significant application of utilities reflective of a higher level of illness severity. 

The most common reason for readmission was sepsis secondary to a biliary source, with nine out of ten most common reasons for readmission being gallstone-related disorders. This indicates that most mortality, morbidity and financial burden were driven by gallstone related disease. Additionally, comparing index to readmission in-hospital mortality rates, a higher percentage of those who were readmitted died compared to index admissions at a significant *p*-value (0.95% vs. 2.55%, *p* = <0.001). Biliary sepsis can occur secondary to etiologies such as acute cholangitis, acute cholecystitis and acute biliary pancreatitis. It is well known that acute cholangitis has a high rate of mortality and has been reported up to 27% in some studies [27]. Although the exact cause of death cannot be determined using our database, it is fair to assume the majority of deaths were due to biliary sepsis from etiologies such as acute cholangitis. This underscores the importance of optimally managing patients to prevent adverse outcomes and the need of system wide changes to remove roadblocks resulting delay in a timely cholecystectomy during index hospitalization.

Between index and readmission hospitalizations comparing mechanical ventilation use (3.64% vs. 25.49%, *p*-value < 0.001), vasopressor use (0.94% vs. 0.29%, *p*-value < 0.001), length of stay (5.16 vs. 4.18, *p*-value < 0.001) and non-routine discharges (35.28% vs. 25.49%, *p* < 0.001) shows significant differences, indicating that readmissions carry higher morbidity, mortality, cost and burden in the health care systems. As biliary sepsis was the most common reason for readmission for this patient population and it is well known that sepsis is linked to higher rates of vasopressor requirements and need for mechanical ventilation. Hence, those readmitted would also be more critically ill and more prone to have non-routine discharges. The decision to withhold cholecystectomy on index admission then would suggest that those readmitted are at risk of worse clinical outcomes with increased morbidity. Patients with comorbidities (adjusted hazard ratio 1.09, *p* < 0.001) have an increased risk of readmission. The disease severity also contributes to readmission rates, however due to limitation of our database, we were not able to evaluate the severity of underlying disease. Further studies on timely cholecystectomy in patients with severe disease and increased comorbidities should be undertaken.

Our analysis of NRD data isolated several independent predictors of 30-day readmission. Independent predictors for 30-day readmission were higher Charlson comorbidity score, younger age (<45 yrs), Medicare and Medicaid insurance, non-routine discharge and longer length of stay. Younger age was associated with increased risk of readmission prior studies looking at heart failure, pneumonia and acute myocardial infarction [28]. Young patients are at increased risk of not having insurance, poor outpatient follow-up and possible increased severity of the disease on presentation. Out of total 887 patients who died during index hospitalization, 694 (78.2%) of them were age ≥65 years-old shows that older patients.

Medicare and Medicaid insurance status were associated with increased emergency department visits and poor outpatient follow-up compared to private insurance, reflecting barriers to outpatient care or lower cost-sharing barriers to emergency department care [29]. Such information may have utility in indicating patients who fit these criteria to better recognize them as having a higher risk of readmission and a potentially worse clinical outcome in readmission. The management of such patients on index admission then could be more geared toward receiving cholecystectomy instead of deferring it. On the other hand, predictors of decreased 30-day readmission studied here could be used as well to stratify patient care and management. These predictive factors that potentially escalate or deescalate management are especially applicable to these hepato–pancreato–biliary disorders as nine out of ten reasons were gallstone-related diseases. 

## 5. Limitations

As with any cross-sectional study, we cannot establish causality but only associations. The National Readmission database, based on ICD-10 coding, can include possible coding errors/ risk factors or diagnoses not entered into the database. The database does not have information on vital signs, lab values and imaging. Because of this, the disease severity could not be assessed, and information on patients who got cholecystectomy as an outpatient after index admission was not obtainable. We could not evaluate social barriers for discharge or readmission, outpatient resource accessibility and medication compliance. NRD does not record out-of-state readmissions. Lastly, we selected patients who did not have cholecystectomy with index hospitalization, leading to selection bias of a sicker group of patients.

## 6. Conclusions

To conclude, patients with gallstone related disease who were discharged without cholecystectomy had a 30-day readmission rate of 12.24% with majority of patients were readmitted with complications of the gallstone-related disease. Patients who were readmitted had significantly higher mortality rates of 2.55% comparing to index hospitalization (0.95%) leading to increased mortality and morbidity. Insurance status, Charlson comorbidity score, non-routine discharge and a longer length of stay are independent predictors of thirty-day readmission. The financial burden incurred from the readmitted patient was $682 million for the year 2018. This study emphasizes the importance of performing cholecystectomy promptly, and the need for further studies on quality measures in gallstone-related disease especially focusing on patients with low socioeconomic status. 

## Figures and Tables

**Figure 1 diseases-09-00089-f001:**
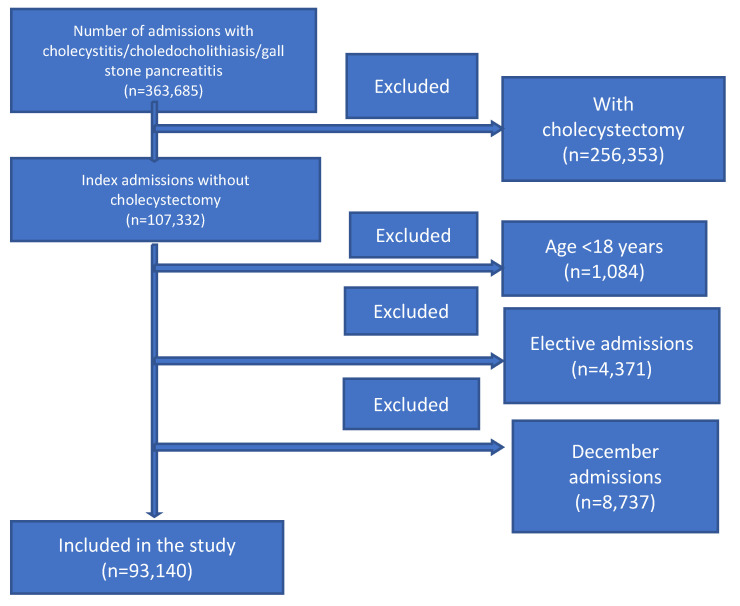
Cholecystectomy inclusion criteria.

**Figure 2 diseases-09-00089-f002:**
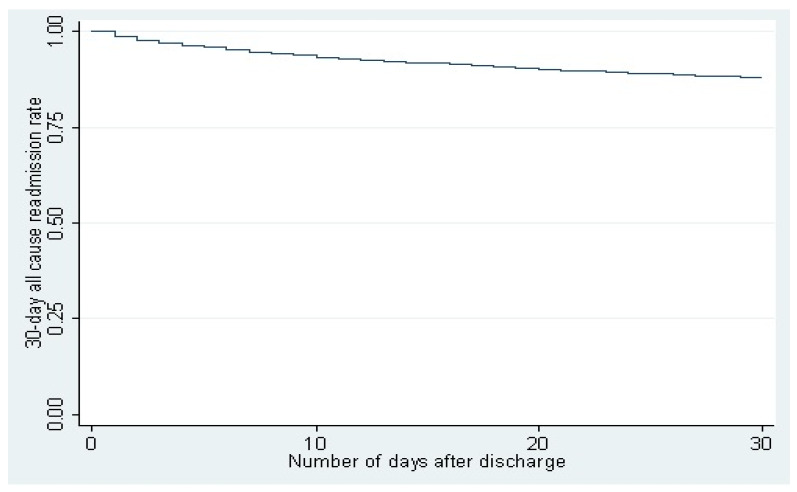
Of 93,140 patients, 92,253 were discharged alive of which 11,292 (12.24%) were readmitted in 30 days. Figure 2 shows Kaplan–Meier survival curve.

**Table 1 diseases-09-00089-t001:** Patient characteristics (*n* = 93,140).

Variable	*n* (%)
Age (years)	
18–44	19.32%
45–64	28.26%
65–84	37.49%
≥85	14.90%
Mean Age (SD)	
Female	61.6 (21.8)
Male	64.9 (17.8)
Sex (Female)	56.44%
Median income in patients zip code ($)	
<$45,999	27.78%
46,000–58,999	28.60%
59,000–78,999	24.19%
>79,000	19.42%
Insurance	
Medicare	55.88%
Medicaid	15.47%
Private	24.27%
Uninsured	4.36%
Charlson comorbidity score	
0	35.28%
1	23.48%
2	14.39%
>3	26.82%
Hospital location	
Large metropolitan area	58.72%
Small metropolitan area	33.98%
Micropolitan area	5.34%
Not metropolitan or micropolitan area	1.95%
Teaching hospital	71.20%
Hospital bed size	
Small	18.31%
Medium	29.01%
Large	52.67%
Mechanical ventilation	1.84%
Vasopressor use	0.29%
Disposition	
Regular	74.51%
Skilled Nursing Facility *	10.01%
Home health *	12.13%
Left against medical advice *	3.35%

* Non-routine discharges include Skilled Nursing Facility, Home health, Left against medical advice.

**Table 2 diseases-09-00089-t002:** Comparison between index admission and readmission.

	Index (93,140)	Readmission (11,292)	*p* Value
Died	887 (0.95%)	288 (2.55%)	<0.001
18–44	3.21%	4.62%	
45–64	18.56%	18.01%	
65–84	45.97%	46.13%	
≥85	32.27%	31.24%	
Mechanical ventilation	1.84%	3.64%	<0.001
Vasopressor use	0.29%	0.94%	<0.001
Length of stay (days)	4.18	5.16	<0.001
Mean total charges ($)	46,078$	61,038$	<0.001
Disposition			<0.001
Regular	74.51%	64.72%	
Skilled Nursing Facility	10.01%	15.43%	
Home health	12.13%	18.10%	
Left against medical advice	3.35%	1.74%	

Total sample size: 93,140; total alive after index admission 92,253; 30-day mortality rate after index hospitalization: 238. Readmit 12.24% (11292); total readmit length of hospital stay burden 57800 days; total readmit charges 682 million $.

**Table 3 diseases-09-00089-t003:** Most common causes of 30-day readmission.

Sepsis, unspecified organism (A419) due to biliary source
2. Calculus of gallbladder with acute cholecystitis without obstruction (K80.00)
3. Biliary acute pancreatitis without necrosis or infection (K85.10)
4. Calculus of gallbladder with acute and chronic cholecystitis without obstruction (K80.12)
5. Acute Cholecystitis (K810)
6. Calculus of gallbladder with chronic cholecystitis without obstruction (K80.10)
7. Acute pancreatitis without necrosis or infection, unspecified (K85.90)
8. Hypertensive heart and chronic kidney disease with heart failure (I13.0)
9. Calculus of gallbladder without cholecystitis without obstruction (K80.20)
10. Calculus of gallbladder without cholangitis or cholecystitis without obstruction (K80.50)

**Table 4 diseases-09-00089-t004:** Predictors for readmission.

Factors	Univariate HR (95% CI)	*p*-Value	Multivariate HR (95% CI)	*p*-Value
Age (years)				
18–44	Reference		Reference	
45–64	1.06 (0.98–1.14)	0.1	0.95 (0.87–1.03)	0.24
65–84	1.07 (0.99–1.15)	0.05	0.8 (0.72–0.9)	<0.001
≥85	0.97 (0.88–1.06)	0.5	0.7 (0.6–0.79)	<0.001
Female	0.86 (0.82–0.91)	<0.001	0.91 (0.87–0.96)	0.002
Median income ($)				
>79,000	Reference		Reference	
59,000–78,999	0.95 (0.88–1.03)	0.3	0.93 (0.85–1.01)	0.12
46,000–58,999	0.97 (0.90–1.05)	0.53	0.94 (0.87–1.02)	0.18
<45,999	1.09 (1.01–1.18)	0.02	1.01 (0.93–1.1)	0.76
Insurance				
Private	Reference		Reference	
Medicare	1.2 (1.18–1.36)	<0.001	1.22 (1.11–1.35)	<0.001
Medicaid	1.4 (1.3–1.53)	<0.001	1.3 (1.2–1.42)	<0.001
Uninsured	1.05 (0.91–1.22)	0.44	0.97 (0.84–1.13)	0.78
Charlson comorbidity score	1.10 (1.09–1.11)	<0.001	1.09 (1.08–1.11)	<0.001
Hospital location				
Large metropolitan area	Reference		Reference	
Small metropolitan area	0.93 (0.88–0.99)	0.03	0.95 (0.89–1.01)	0.12
Micropolitan area	1.01(0.88–1.15)	0.84	0.91 (0.78–1.07)	0.28
Not metropolitan or micropolitan area	0.95 (0.77–1.16)	0.65	0.96 (0.76–1.22)	0.79
Teaching hospital	0.97 (0.92–1.03)	0.44	0.96 (0.9–1.03)	0.34
Hospital bed size				
Small	Reference		Reference	
Medium	0.95 (0.87–1.03)	0.24	0.94 (0.86–1.02)	0.19
Large	1.03 (0.96–1.11)	0.33	1.02 (0.94–1.11)	0.52
Mechanical ventilation	1.27 (1.05–1.54)	0.01	0.88 (0.71–1.1)	0.28
Vasopressor use	1.09 (0.64–1.88)	0.73	0.64 (0.36–1.15)	0.14
Disposition				
Regular	Reference		Reference	
Skilled nursing facility	1.36 (1.25–1.48)	<0.001	1.19 (1.07–1.32)	0.001
Home health	1.37 (1.27–1.47)	<0.001	1.22 (1.12–1.33)	<0.001
Left against medical advice	2.15 (1.9–2.42)	<0.001	2.17 (1.92–2.46)	<0.001
Length of stay	1.01 (1.01–1.02)	<0.001	1.01 (1.01–1.02)	<0.001

## Data Availability

Dataset is available at https://www.hcup-us.ahrq.gov/db/nation/nrd/Introduction_NRD_2010-2018.jsp, accessed on 2 December 2021.

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
