# Peer review of "Cutting out Cholecystectomy on Index Hospitalization Leads to Increased Readmission Rates, Morbidity, Mortality and Cost"

_diseases, 2021, doi:10.3390/diseases9040089_

Round 1
Reviewer 1 Report
Dear Author,
I have read with interest your manuscript dealing with 30d readmission rate after biliary stone disease events.
I really appreciate the aim of the study, however, I found several issues that limit the overall quality of the study.
In detail:
- several prospective studies, RCTs, and even meta-analysis aimed to compare the best timing for cholecystectomy; this study does not add any relevant evidence in the field;
- the study design is weak: in particular, the authors censored all patients admitted on December since it was impossible to accurately evaluate 30d readmission.
Finally, I am not able to read the reference list; I found only 10 reference in a setting, biliary stone disease, with a large amount of robust evidence.
Author Response
Reviewer: 1
Reviewer's report:
We thank the reviewer for his comments
Comment #1:
-
several prospective studies, RCTs, and even meta-analysis aimed to compare the best timing for cholecystectomy; this study does not add any relevant evidence in the field;
The importance of this study is that patients who do not undergo cholecystectomy during the index hospitalization, may not undergo one as an outpatient in a timely manner and may be readmitted as a consequence of this. Our study shows that these patients are readmitted due to sepsis secondary to a biliary source and this suggests that they have not undergone an outpatient cholecystectomy. Hence, we need to identify patients who are at high risk for failure of outpatient cholecystectomy and prioritize them for a cholecystectomy during the index hospitalization.
Comment #2:
-
the study design is weak: in particular, the authors censored all patients admitted on December since it was impossible to accurately evaluate 30d readmission.
As this data involves the national readmission database and given the nature of this dataset, we were unable to account for the month of December. For patients who are admitted in December and their hospitalization extends into January of the following year, it is not possible to identify the portion of the hospitalization of December and link it to that of January as one index hospitalization. As the dataset does not provide any link between December of the previous year and the January of the following year. Similarly, a patient hospitalized and discharged in December and then readmitted in January is identified by the dataset as two separate patients and not the same individual. This is an inherent limitation of the database and this unfortunately limits us from including the month of December as 30-day readmission rates cannot be calculated for this month. Many studies that utilize this database also do not include the month of December. The following is an example of such a study:
Siva Harsha Yedlapati, Scott H Stewart, Predictors of Alcohol Withdrawal Readmissions, Alcohol and Alcoholism, Volume 53, Issue 4, July 2018, Pages 448–452, https://doi.org/10.1093/alcalc/agy024
Comment #3:
Finally, I am not able to read the reference list; I found only 10 reference in a setting, biliary stone disease, with a large amount of robust evidence.
We agree with the reviewer. We have updated the paper by incorporating more references into the study to support our findings
Top of Form
Review Report 2
Open Review
(x) I would not like to sign my review report
( ) I would like to sign my review report
English language and style
( ) Extensive editing of English language and style required
(x) Moderate English changes required
( ) English language and style are fine/minor spell check required
( ) I don't feel qualified to judge about the English language and style
|
|
|
|
Yes |
Can be improved |
Must be improved |
Not applicable |
|
Does the introduction provide sufficient background and include all relevant references? |
(x) |
( ) |
( ) |
( ) |
|
Is the research design appropriate? |
( ) |
( ) |
( ) |
(x) |
|
Are the methods adequately described? |
(x) |
( ) |
( ) |
( ) |
|
Are the results clearly presented? |
(x) |
( ) |
( ) |
( ) |
|
Are the conclusions supported by the results? |
(x) |
( ) |
( ) |
( ) |
Comments and Suggestions for Authors
There is extensive existing literature around the debate between early vs. delayed cholecystectomy for gallstone disease.
However, despite it being a retrospective observational study I found your article interesting and relevant. It reinforces the message (especially in the midst of a pandemic) of considering index admission cholecystectomies to prevent later morbidity and mortality.
I feel that your article could be improved by addressing style and grammar throughout the manuscript and especially where you have listed limitations and conclusions.
Reviewer 2 Report
There is extensive existing literature around the debate between early vs. delayed cholecystectomy for gallstone disease.
However, despite it being a retrospective observational study I found your article interesting and relevant. It reinforces the message (especially in the midst of a pandemic) of considering index admission cholecystectomies to prevent later morbidity and mortality.
I feel that your article could be improved by addressing style and grammar throughout the manuscript and especially where you have listed limitations and conclusions.
Author Response
Reviewer: 2
Reviewer's report:
There is extensive existing literature around the debate between early vs. delayed cholecystectomy for gallstone disease.
However, despite it being a retrospective observational study I found your article interesting and relevant. It reinforces the message (especially in the midst of a pandemic) of considering index admission cholecystectomies to prevent later morbidity and mortality.
I feel that your article could be improved by addressing style and grammar throughout the manuscript and especially where you have listed limitations and conclusions.
We thank the reviewer for their comments.
We agree with reviewer and addressed the style and grammar throughout the manuscript. We have updated the limitations and conclusion section accordingly.
Reviewer 3 Report
In this manuscript, the authors showed that readmitted patients had higher mortality rate, intubation rate, vasopressor use and had non-routine discharge when compare to index admission for the patients with a diagnosis of cholelithiasis or acute biliary pancreatitis. The result was interesting. However, I advise the authors to make some modifications in the manuscript.
Major points
#1. In the results section, the authors described that “A total of 887 (0.95%) died during index hospitalization, while 288 (2.57%) died during readmission (0.95% vs 2.57%, p <0.001) “. Why mortality rate was higher in readmitted patients? I think the authors should describe cause of death. Was cause of death all related to hepatopancreatobiliary disorders? Please discuss this point in the Discussion section and if possible, please show “cause of death” in the Result section.
#2. In the results section, the authors described that readmitted patients had higher mechanical ventilation (3.64% vs 1.84% p <0.001), vasopressor use (0.94% vs 0.29% p<0.001) and non-routine discharge (35.28% vs 25.49% p<0.001). I also want to know the reason of these results. Were these results all related to hepatopancreatobiliary disorders? Or due to the patients bad general condition? Please discuss this point in the Discussion section.
#3. In the results section, the authors described about independent predictors of readmission. I want to know why younger age (<45 yrs) and Medicare or Medicaid insurance were independent predictors of readmission. Was medical insurance affect the treatment policy of the patients? Please discuss these points in the Discussion section.
Minor points
#1. Could the authors explain about “non routine discharge”?
Author Response
Reviewer:3
We thank the reviewer for his comments.
Comment #1:
In the results section, the authors described that “A total of 887 (0.95%) died during index hospitalization, while 288 (2.57%) died during readmission (0.95% vs 2.57%, p <0.001) “. Why mortality rate was higher in readmitted patients? I think the authors should describe cause of death. Was cause of death all related to hepatopancreatobiliary disorders? Please discuss this point in the Discussion section and if possible, please show “cause of death” in the Result section.
Updated discussion reads as follows:
The most common etiology for readmission based on our study was sepsis due to a biliary source, with nine out of ten most common reasons for readmission being gallstone-related disorders. As sepsis is associated with a high risk for mortality, it is likely that this accounted for the higher mortality rate in those who were readmitted. This has been explained further in our manuscript.
Comment #2:
In the results section, the authors described that readmitted patients had higher mechanical ventilation (3.64% vs 1.84% p <0.001), vasopressor use (0.94% vs 0.29% p<0.001) and non-routine discharge (35.28% vs 25.49% p<0.001). I also want to know the reason of these results. Were these results all related to hepatopancreatobiliary disorders? Or due to the patients bad general condition? Please discuss this point in the Discussion section.
As biliary sepsis was the most common etiology for readmission and as sepsis is associated with need for vasopressor support and mechanical ventilation, it is speculated that this is the likely rationale for these findings. As patients having such complex needs tend to be more critically ill, these patients have more complicated hospitalizations and more needs on discharge. They often require placement to nursing facilities or rehabilitation centers (non-routine discharges). This has been elaborated in the discussion section.
Comment #3:
#3. In the results section, the authors described about independent predictors of readmission. I want to know why younger age (<45 yrs) and Medicare or Medicaid insurance were independent predictors of readmission. Was medical insurance affect the treatment policy of the patients? Please discuss these points in the Discussion section.
Our study revealed that patients aged 65 and above tended to have a higher mortality rate on the index hospitalization as compared to the younger patients and this reflects in the overall lower readmission rates. Also younger patients may have poor outpatient follow up and this can contribute to the relatively higher readmission rates. Medicare and Medicaid is associated with poor socioeconomic status and this will contribute to risk for poor follow up with outpatient cholecystectomy and hence higher readmission rates. This has been explained in the text as well.
Minor points
#1. Could the authors explain about “non routine discharge”?
Non routine discharges include discharges to a skilled nursing facility or home health or left against medical advice